# Impending Low Intake Dehydration at Admission to A Geriatric Ward- Prevalence and Correlates in a Cross-Sectional Study

**DOI:** 10.3390/nu12020398

**Published:** 2020-02-02

**Authors:** Zyta Beata Wojszel

**Affiliations:** 1Department of Geriatrics, Medical University of Bialystok, Fabryczna 27, 15-471 Bialystok, Poland; wojszel@umb.edu.pl; Tel.: +48-85-869-4982; 2Department of Geriatrics, Hospital of the Ministry of Interior and Administration in Bialystok, Fabryczna 27, 15-471 Bialystok, Poland

**Keywords:** fluid intake, water intake, impending dehydration, geriatric inpatients, frailty, procognitive medications, acetylcholine esterase inhibitor, chronic renal disease, electrolyte, osmolyte

## Abstract

Dehydration risk increases with frailty and functional dependency, but a limited number of studies have evaluated this association in hospitalized geriatric patients. This cross-sectional study aimed to assess the prevalence and determinants of dehydration in patients admitted to the geriatric ward. Dehydration was diagnosed when calculated osmolarity was above 295 mMol/L. Logistic regression analyses (direct and stepwise backward) were used to assess determinants of impending dehydration. 358 of 416 hospitalized patients (86.1%) were included: 274 (76.5%) women, and 309 (86.4%) 75+ year-old. Dehydration was diagnosed in 209 (58.4%) cases. Significantly higher odds for impending dehydration were observed only for chronic kidney disease with trends for diabetes and procognitive medication when controlling for several health, biochemical, and nutritional parameters and medications. After adjusting for “dementia” the negative effect of “taking procognitive medications” became a significant one. Chronic kidney disease, diabetes, taking procognitive medications and hypertension were the main variables for the outcome prediction according to the stepwise backward regression analysis. This may indicate an additional benefit of reducing the risk of dehydration when using procognitive drugs in older patients with dementia.

## 1. Introduction

Different factors make older people especially prone to dehydration, and old age is pointed as an important determinant of inadequate hydration in hospitalized patients [1,2]. Very frequently, older adults do not perceive water as an essential nutrient and additionally, many intrinsic and extrinsic factors are connected with the increased dehydration risk in this population. The former ones include age-related physiological changes (decline in intracellular water and fluid reserves, changes in renal function, resulting in the inability to concentrate urine, altered thirst perception), diseases combined with increased fluid loses because of vomiting or diarrhea, and changes in the functional status of an older person (such as cognitive disability or mobility impairment resulting in restricted fluid intake). The latter ones include, among others, medications adverse effects, insufficient social support network in community-dwelling older people or inadequate staffing in caring institutions [3,4,5,6]. 

Inadequate hydration can contribute to increased morbidity in older adults. Unfavorable consequences of dehydration include, among others, the increased risk of delirium [7,8], aspiration pneumonia [9], acute kidney failure or urinary tract infections [10], falls and fractures [11], and result in greater odds of death [12]. Despite the general awareness of the importance of this problem among medical staff, very often, older patients who are dehydrated are not being identified, or their hydration status is misdiagnosed [13,14]. Making a proper diagnosis is often a challenge, mainly because signs and symptoms that are usually ascribed to dehydration have debatable diagnostic value in the older population [15,16]. Untreated dehydration negatively affects health and wellbeing, and significantly increases mortality rates of hospitalized older patients [2]. This is an independent factor increasing health care expenditures connected with an increased utilization of intensive care units, short- and long-term care facilities, and hospital readmission rates [17].

Although serum osmolality measured with the use of standardized analytic procedures, with a cut-off of 301 ± 5 mmol/kg possessing the best diagnostic accuracy, is likely to be the best indicator of static dehydration [18], its direct measurement is not routinely undertaken in Polish hospitals, due to costs, among others. In recent years, a two-stage diagnostic process of screening for dehydration in older patients was proposed to be used in clinical practice. As the first step, serum calculated osmolarity should be assessed routinely. For those identified as being at high risk (with impending dehydration; it means when calculated serum osmolarity is above 295 mMol/L) serum osmolality measurement should be performed [16]. Calculated osmolarity approximates osmolality. It was proven that if it was calculated with the Khajuria and Krahn equation from routine clinical biochemical variables (serum sodium, potassium, urea, and glucose values), then it would be best able to predict measured osmolality in older people [19], independently of diabetes status [20]. This two-stage routine, accepted in the latest European Society for Clinical Nutrition and Metabolism (ESPEN) guideline on nutrition in geriatrics [21], can help to identify older patients that need intervention and improve their hydration in a time- and cost-saving manner. 

Although it is indicated that dehydration risk increases with frailty and functional dependency, there is a limited number of research studies that evaluate this association in hospitalized older adults. Thus, the aim of this study was to examine the prevalence of impending dehydration (assessed with calculated osmolarity) in patients admitted to the geriatric ward, and its association with health and functional ability characteristics, as well as with patients’ frailty status. 

## 2. Materials and Methods 

Data collected during the cross-sectional study on frailty and multimorbidity in patients of the Department of Geriatrics (Hospital of the Ministry of Interior and Administration in Bialystok, Poland) [22,23] were subjected to a secondary analysis. All patients admitted to the ward during 7 months at the turn of 2014 and 2015, with serum sodium, potassium, glucose and urea values determined on admission, were included in the analysis. Osmolarity was calculated with Khajuria and Krahn equation (osmolarity = 1.86 × (Na^+^ + K^+^) + 1.15 × glucose + urea + 14; each component measured in mmol/L) [24]. 

Database included information on patient’s age, gender, place of residence, history of hospitalizations and incidents of falls in the previous year, prevalence of 15 chronic diseases (atrial fibrillation, chronic arthritis, chronic cardiac failure, chronic obstructive pulmonary disease, chronic renal disease, dementia, diabetes/ prediabetes, hypertension, ischemic heart disease, myocardial infarction, neoplasm, osteoporosis, parkinsonism, peripheral arterial disease, and stroke), and medications taken before hospitalization. The ability to perform activities of daily living (ADL) was assessed with the Barthel Index [25] and 6 instrumental ADL items of Duke OARS scale- IADL [26]. It also included the results of other tests and scales, collected within comprehensive geriatric assessment performed in the department: the Abbreviated Mental Test Score (AMTS) [27] (the assessment of cognitive abilities), Geriatric Depression Scale (GDS) [28] (the evaluation of patient’s emotional health), the Performance Oriented Mobility Assessment (POMA) [29] and the Timed Up and Go test (TUG) [30] (the assessment of the risk of falls), and the Norton Scale (the risk of pressure sores) [31]. Nutritional health was evaluated with body mass index (BMI), waist circumference, waist-hip ratio (WHR), and the Mini Nutritional Assessment-Short Form (MNA-SF) [32]. Data on serum sodium, potassium, calcium, urea, creatinine, fasting glucose, hemoglobin and albumin were collected. The glomerular filtration rate (GFR) was calculated with the CKD-EPI formula [33]. Gait speed was evaluated during the 4.57 m walk at usual pace and hand grip strength of the dominant hand was measured with a manual hydraulic dynamometer SAEHAN DHD-1 (mean of two results). The Seven-Item Canadian Study of Health and Aging Clinical Frailty Scale (CFS) was used to evaluate frailty status [34]. Blood pressure records at admission and prevalence of orthostatic hypotension assessed with active standing test were checked.

### 2.1. Study Parameters

Impending low-intake dehydration was diagnosed if the calculated osmolarity was above 295 mMol/L (“dehydration+” group) [19]. Patients who did not fulfil this criterion were included in the “dehydration-“group. Therefore, it should be underlined that the term “dehydration” corresponds to “impending dehydration” in this study, as patients were classified based on calculated osmolarity, and not on plasma osmolality measurement results.

Hyponatremia was diagnosed if serum sodium was below 135 mmol/L, and hypernatremia if it was above 145 mmol/L. The score of 6 or 7 of CFS was classified as severe frailty. Serum albumin <35 g/L, and MNA-SF score below 8 pointed to malnutrition. Abdominal obesity was defined as a waist circumference value above 0.90 m for men and above 0.85 m for women. If the active standing test in the first or the third minute showed a drop in systolic pressure (at least by 20 mm Hg) and/or in diastolic pressure (at least by 10 mmHg), orthostatic hypotension was diagnosed. If the serum hemoglobin was below 8.7 mmol/L in men and below 7.5 mmol/L in women, anemia was diagnosed. GFR <60 mL/min/1.73 m^2^ was used as the criterion of the chronic kidney disease (CKD). POMA score <19 was treated as suggestive for the high risk of falls. Sarcopenia was diagnosed in men if grip strength was lower than 27 kg, and in women if it was lower than 16 kg [35]. This could be classified as severe sarcopenia based on physical performance: if it was accompanied by a gait speed equal or lower than 0.8 m/s, and/or TUG equal or higher than 20 s. Five or more drugs taken was treated as “polypharmacy”, and five or more diseases of 15 listed- as “multimorbidity”.

### 2.2. Statistical Analysis

Analyses were done with IBM SPSS Version 18 Software suit (SPSS, Chicago, IL, USA). Variables’ distribution was checked with Shapiro–Wilk tests. They were presented as frequency and percentage (categorical variables), as means and standard deviation (normally distributed quantitative variables), or as medians and interquartile range (not normally distributed quantitative variables). Proportions were compared using χ2 tests, while the Student’s *t*-test for independent samples and Mann–Whitney U test were used to compare means and medians. Odds ratios with 95% confidence interval (CI) and *p* value were estimated by multivariable logistic regression methods (direct and stepwise backward) with impending dehydration as outcome variable. Dehydration predictors with a *p* value lower than 0.1, without those highly correlated to avoid multicollinearity effect, were included. In all analyses, a two-tailed *p* value of less than 0.05 was regarded as significant. Analyses did not include missing data.

### 2.3. Ethics Approval

The source study was approved by the Ethics Committee at Medical University of Bialystok (no R-I-002/305/2013). All the procedures performed in the study were in accordance with the ethical standards of the Medical University of Bialystok research committee and with the Helsinki declaration and its later amendments. The study can be classified as a study of ‘usual practice’. All study participants gave their informed consent to participate in it.

## 3. Results

Plasma osmolarity was calculated in 358 (86.1%) of 416 patients admitted to the geriatric department, and they were included in the analysis (Figure 1). Most of them were at the age of 75 or more (86.3%), female (76.5%), and community dwelling (96.9%). The median value of calculated osmolarity in the whole group was 296.5 mMol/L (IQR, 292.2–299.6 mMol/L). In 209 (58.4%) cases, a threshold of impending dehydration (295 mMol/L) was exceeded. In the “dehydration+” group, significantly higher median values of sodium (141 mmol/L; IQR, 140–142 mmol/L versus 138 mmol/L; IQR, 136–139 mmol/L, *p* < 0.001), urea (7.8 mmol/L; IQR, 6.2–9.9 mmol/L versus 5.8 mmol/L; IQR, 4.9–7.3 mmol/L, *p* < 0.001), and fasting glucose (5.6 mmol/L; IQR, 5.2–6.7 mmol/L versus 5.3 mmol/L; IQR, 4.9–5.8 mmol/L, *p* < 0.001), and mean value of potassium (4.50mmol/L; SD,0.46mmol/L versus 4.33; SD, 0.44mmol/L, *p* = 0.001) were observed. Hyponatremia was diagnosed in 6.7% of cases, significantly more often in the “dehydration-“ group (13.4% versus 1.9%, *p* < 0.001). No cases of hypernatremia were diagnosed in the studied group.

The “dehydration+” group and patients with proper hydration did not differ in age, sex, in percentage of people living alone or in a caring institution, and in many other analyzed characteristics. They differed significantly in the median number of chronic diseases (5; IQR, 4–6 versus 4; IQR, 3–6 in the “dehydration-“ group, *p* = 0.001) and the number of drugs taken (7; IQR, 5–10 versus 7; IQR, 4–9 in “dehydration-“ group, *p* = 0.01). In dehydrated patients, significantly higher percentages of multimorbidity (64.1% versus 51.0% in the “dehydrated-“ group, *p* = 0.02) and polypharmacy (83.5% versus 71.5% in the “dehydrated-“ group, *p* = 0.008) were noticed (Table 1). Moreover, a significantly higher prevalence of hypertension (84.2% versus 68.5%, *p* = 0.001), diabetes (36.4% versus 21.5%, *p* = 0.002) and chronic kidney disease (63.6% versus 38.9%, *p* < 0.001) in the “dehydration+” group were observed. Inadequately hydrated patients significantly more frequently were on diuretics (53.7% versus 38.9%, *p* = 0.009), drugs affecting the renin-angiotensin-aldosterone system (67.3% versus 56.3%, *p* = 0.04), beta-blockers (68.8% versus 56.9%, *p* = 0.03), lipid-lowering medications (38.0% versus 27.1%, *p* = 0.04; in all, apart 1 case on fibrate, patients were on statins), but less frequently on procognitive medications- acetylcholinesterase inhibitors (AChE-Is) and/or memantine (9.8% versus 17.4%, *p* = 0.05). 

The “dehydration+” group and “dehydration-“ group did not differ in the majority of characteristics of functional health. It concerned the Barthel Index, IADL, and GDS scores, handgrip strength, gait speed, TUG result (and their values pointing to sarcopenia and severe sarcopenia), POMA score and increased risk of falls, WHR and percentage of patients with abdominal obesity, Norton scale score, median value of CFS and percentage of patients classified as severely frail. In the “dehydration+” group significantly higher- comparing to non-dehydrated patients- values of some parameters characterizing patient’s nutritional health, such as mean BMI (29.8 kg/m^2^; SD, 6.2 kg/m^2^ versus 28.1 kg/m^2^; SD, 5.6 kg/m^2^, *p* = 0.01), the percentage of patients with a BMI > 30 kg/m^2^ (46.3% versus 33.6%, *p* = 0.03), and waist circumference median value (0.98 m; IQR, 0.87–1.09 m versus 0.94 m; 0.85–1.04 m, *p* = 0.04), were observed (Table 2). 

A direct multivariable logistic regression analysis was carried out on impending dehydration as outcome and 14 predictors: hypertension, ischemic heart disease, cardiac heart failure, diabetes/ prediabetes, chronic kidney disease, taking procognitive medications, beta-blockers, statins, selective serotonin reuptake inhibitors (SSRI), diuretics, ACE-I or ARB, abdominal obesity, BMI > 30 kg/m^2^, and polypharmacy (Table 3). A few variables meeting the criterion *p* < 0.1 (taking memantine, taking AChE-Is, BMI, waist circumference, multimorbidity, number of diseases, and number of drugs taken) were not included in the logistic regression after testing for correlation with other variables and for their multicollinearity effect. Significantly higher odds for calculated osmolarity >295 mMol/L were observed only for chronic renal disease (odds ratio, 2.40; 95% CI, 1.35–4.10; *p* = 0.003). There was also a trend for an association with variable “diabetes” (odds ratio, 1.80; 95% CI, 0.96–3.37; *p* = 0.07), and with “taking procognitive medications” (odds ratio, 0.48; 95% CI, 0.21–1.10; *p* = 0.08), when controlling for hypertension, ischemic heart disease, cardiac heart failure, chronic kidney disease, taking beta-blockers, lipid lowering medications, SSRI, diuretics, ACE or ARB, abdominal obesity, BMI > 30 kg/m^2^, and polypharmacy (Model 1). After adjusting for the variable “dementia” (despite of its not significant association with impending dehydration in bivariate analysis), the odds connected with “taking procognitive medications” became a significant one (odds ratio, 0.37; 95% CI, 0.14–0.99; *p* = 0.048)- Model 2. For both models, an overall prediction success rate of 65.7% was observed, with 78.7% of the “dehydration+” status (sensitivity) and 47.3% of “dehydration-“ status (specificity) correctly predicted. The adjustment for age and gender, as biologically important covariates (Model 3), did not influence the results. The prediction success was similar, with 78.1% of “dehydration+” cases and 46.4% of “dehydration–“ cases correctly predicted, for an overall success rate of 64.9%. 

A stepwise backward logistic regression analysis was performed on impending dehydration at admittance to the department as outcome and 17 predictors constructing Model 4 (Table 4). The results of the analysis suggested the model with 4 variables only: chronic kidney disease (odds ratio, 2.53; 95% CI, 1.51–4.27; *p* < 0.001), diabetes (odds ratio, 1.9; 95% CI, 1.04–3.44; *p* = 0.04), taking procognitive medications (odds ratio, 0.42; 95% CI, 0.20–0.88; *p* = 0.02), and hypertension (odds ratio, 1.75; 95% CI, 0.91–3.34; *p* = 0.09). Prediction success was the best in the case of Model 4, with 72.3% of “dehydration+” cases and 59.1% of “dehydration–“ cases correctly predicted, for an overall success rate of 66.8%.

## 4. Discussion

Water homeostasis is crucial for life. It is an essential body nutrient, although very often, it is not seen as such. Maintaining normal hydration in older patients is hindered by a number of factors [4]; therefore, plasma hypertonicity and dehydration is observed more frequently in this population than in the younger adults. Mild and overt hypertonicity were observed respectively in 40% and 20% of the community-dwelling (aged 20 to 90 years) sample of the Third National Health and Nutrition Examination Survey, and it positively correlated with more advanced age and decreased water consumption in older adults [36]. In the United States, dehydration was listed as one of the five reported diagnoses in 6.7% of Medicare hospitalizations (patients in the age of 65 years or more) [2]. The results of the completed study reveal that impeding dehydration status, assessed with plasma osmolarity greater than 295 mMol/L, affected a high proportion of patients at admission to the geriatric department. It was diagnosed in 58.4% of patients, and its prevalence was comparable to those observed in long-term care residents rather [37,38]. It can be explained by the high prevalence of different characteristics making this group more prone to dehydration. The study group was highly burdened with psycho-physical disability and dependent on others. Very frequently study participants were affected by multimorbidity (58.7%), polypharmacy (78.6%), dementia (34.1%), sarcopenia (46.2%), and frailty (in 25.4% it was a severe frailty according to CFS). Because of that, albeit the old age is pointed to be an important determinant of dehydration status in hospitalized patients, the advancement of age was not connected with the higher risk of impending dehydration in the conducted study. Calculated osmolarity higher than 295 mMol/L was significantly more frequently observed in patients with multimorbidity and with polypharmacy. The results were similar to those obtained by Lavizzo-Mourey et al. among acutely ill elderly nursing home residents [39].

According to some studies, plasma hypertonicity might be an indicator of frailty starting to develop in nondisabled older adults, and a predictor of disability and mortality in this population [40]. Despite this, no association between impeding dehydration and such functional parameters as Barthel Index score, TUG test result, POMA, GDS score, gait speed, sarcopenia assessed with grip strength, falls in the last 12 months, Norton scale score, median value of CFS result, or percentage of patients qualified as severely frail, was observed. This may also be the result of the general characteristics of the study population.

Patients with a calculated osmolarity higher than 295 mMol/L had a significantly higher mean value of BMI. Impeding dehydration was also significantly more frequently observed in patients with BMI > 30 kg/m^2^, and with higher waist circumference in the study. This may be due to the altered fluid distribution in obesity reflecting plasma hypertonicity [41]. Obesity is also connected with a higher risk of chronic kidney disease [42].

Based on Model 4, it can be concluded that chronic kidney disease, diabetes, taking procognitive medications and hypertension turned out to be the main variables for the outcome- impending dehydration- prediction. The results of the presented analysis are consistent with findings of DRIE’s study conducted in the United Kingdom, suggesting that—apart from cognitive abilities—diabetes status and kidney function are connected with dehydration status in frail care homes residents [38]. Aging is a risk factor for kidney function deterioration, but also both hypertension and diabetes are connected with an increased risk of nephropathy, regardless of age [43], leading to metabolic and electrolyte homeostasis disturbances, and increased risk of dehydration. Increased arterial pressure is also mentioned among inhibitors of thirst and arginine vasopressin release [4]. High blood glucose levels lead to osmotic diuresis and decreased hydration in the body. Hyperglycemic dehydration syndrome is one of the important reasons for hospitalizations among older type 2 diabetes patients [44]. The proper treatment of diabetes should be a priority in the above case, and—in parallel with this—water and electrolyte deficiencies should be supplemented. 

Many medications taken by older individuals can influence water homeostasis. Drugs taken at admittance, significantly more frequently reported in the “dehydration+” group included diuretics, ACE-Is/ARBs, beta-blockers, and statins. Diuretics force sodium wasting and reduce the diluting capacity of the kidney [45]. Inhibition of angiotensin-converting enzyme can influence the natriuretic and diuretic action of atrial natriuretic peptide [46]. Correlation with drugs other than diuretics may, however, be indirectly due to the increased frequency of their use in hypertension. Medications reported less frequently in the “dehydration+” group included procognitive medications and SSRIs (in case of both groups on the verge of significance). This can be explained, for instance, with SIADH, that can be caused by psychotropic medications, and antidepressiva (including SSRI), among others [47]. But there was no significant correlation between these classes of medications and hyponatremia observed (data not presented). It is rather about the positive impact of SSRIs and procognitive drugs on cognitive functions and motivation, that can improve older patients’ ability to efficaciously communicate their needs, decreased in dementia and depression. In multivariable logistic regression analysis only procognitive medications revealed to be independently connected with impending dehydration, when the influence of dementia was controlled for (the direction of correlation was an opposite one). It is also worth mentioning that acetylcholine is involved in the sensation of thirst activation. Acetylcholine level in the brain is decreasing with aging, and down-regulation of alpha 7 nicotinic acetylcholine receptors is observed [48], making the brain less responsive to acetylcholine. Maybe AChE-Is can also influence this mechanism in dementia patients. The research published recently by Bialecka-Debek and Pietruszka did not confirm the relationship between the level of cognitive abilities and dehydration estimated based on the specific gravity of urine, but this study was carried out on a very small sample of older volunteers, with normal or mildly impaired cognitive functions, without people with dementia [49]. In addition, the reliability of the diagnosis of dehydration based on the specific gravity of urine in older patients is at least debatable, and it is not recommended now [21]. The results of the current analysis are rather consistent with the findings of Hooper’s at al. study that confirmed the higher odds for dehydration in the case of demented residents in residential care [38].

The study has limitations that ought to be emphasized. First of all, as is underlined above, the term “dehydration” used in the manuscript corresponds to “impending dehydration” in this study (its diagnosis was based on calculated osmolarity). Plasma osmolality measurement should have been performed in the “dehydration+” group to confirm this diagnosis. The results of the study can be generalized for the population with similar characteristics and for similar institutions only, and not for the whole population of older patients. Additionally, the retrospective design of the study, connected with the secondary analysis of previously collected data, resulted in the non-availability of some pieces of information (as indicated in the tables).

## 5. Conclusions

The study results reveal the high prevalence of impending dehydration in older patients in the geriatric ward—above half of them had a calculated osmolarity higher than 295 mMol/L at admittance. Chronic kidney disease, diabetes and hypertension were the main independent predictors connected with significantly higher odds for dehydration status, and taking procognitive medications status proved to be related with significantly lower odds for dehydration after adjustment for diagnosed dementia. This may point to the additional benefit of using procognitive drugs in people with cognitive disability.

## Figures and Tables

**Figure 1 nutrients-12-00398-f001:**
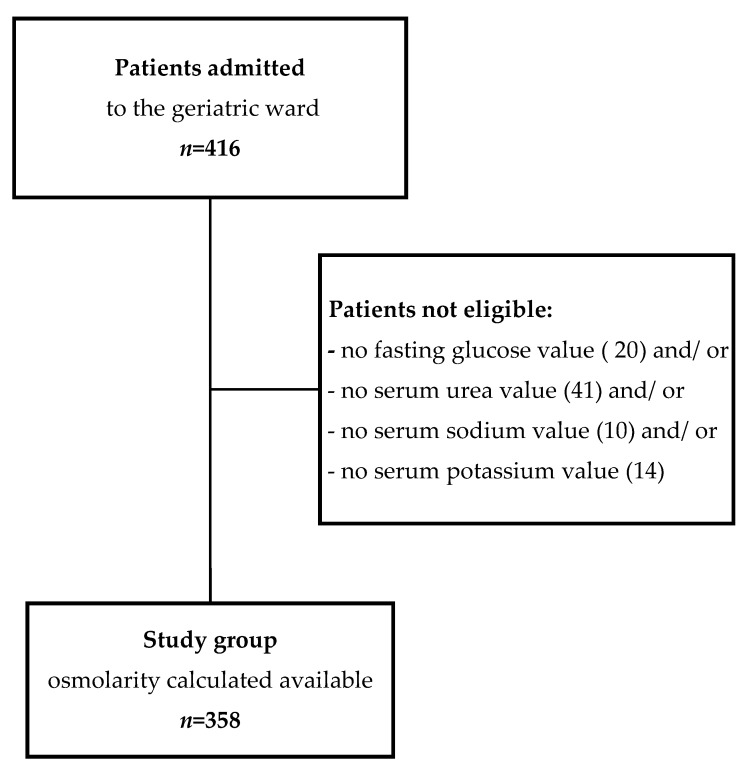
Flow chart of patient enrollment.

**Table 1 nutrients-12-00398-t001:** Characteristics of participants by dehydration status- sociodemographic, health, biochemical parameters and medications.

	All	Dehydration+	Dehydration-	*p* ^1^	Missing Values
*n* (%)	358 (100.0)	209 (58.4)	149 (41.6)		
Sociodemographic parameters					
Age (y), Mdn (IQR)	82 (78–86)	82 (78–86)	82 (77–86)	0.33	
Age (75+), *n* (%)	309 (86.3)	185 (88.5)	124 (83.2)	0.16	-
Sex (F), *n* (%)	274 (76.5)	161 (77.0)	113 (75.8)	0.80	-
Residence (rural), *n* (%)	77 (21.5)	45 (21.5)	32 (21.59)	1.00	-
Living in long term care, *n* (%)	11 (3.1)	7 (3.3)	4 (2.7)	1.0	-
Living alone, *n* (%)	105 (30.3)	64 (31.7)	41 (28.3)	0.55	11
Health parameters					
Hospitalizations in the last year, *n* (%)	98 (27.6)	55 (26.6)	43 (29.1)	0.63	3
Chronic diseases, Mdn (IQR)	5.0 (3.0–6.0)	5.0 (4.0–6.0)	4.0 (3.0–6.0)	**0.001**	
Multimorbidity, *n* (%)	210 (58.7)	134 (64.1)	76 (51.0)	**0.02**	-
Parkinson disease, *n* (%)	51 (14.2)	29 (13.9)	22 (14.8)	0.88	-
Dementia, *n* (%)	122 (34.1)	69 (33.0)	53 (35.6)	0.65	-
Depression, *n* (%)	154 (55.8)	87 (53.0)	67 (59.8)	0.27	82
Ischemic heart disease, *n* (%)	191 (53.4)	121 (57.9)	70 (47.0)	0.05	-
Chronic cardiac failure, *n* (%)	141 (39.4)	90 (43.1)	51 (34.2)	0.10	-
NYHA class I/II, *n* (%)	70 (49.6)	44 (48.9)	26 (51.0)	0.86	
NYHA class III/IV, *n* (%)	71 (50.4)	46 (51.1)	25 (49.0)		
Atrial fibrillation, *n* (%)	88 (24.6)	52 (24.9)	36 (24.2)	0.90	-
Myocardial infarction, *n* (%)	35 (9.8)	24 (11.5)	11 (7.4)	0.21	-
Peripheral arterial disease, *n* (%)	57 (15.9)	32 (15.3)	25 (16.8)	0.77	-
Diabetes, *n* (%)	108 (30.2)	76 (36.4)	32 (21.5)	**0.002**	-
Hypertension, *n* (%)	278 (77.7)	176 (84.2)	102 (68.5)	**0.001**	-
Stroke/ TIA, *n* (%)	52 (14.5)	30 (14.4)	22 (14.8)	1.00	-
Neoplasm, *n* (%)	31 (8.7)	16 (7.7)	15 (10.1)	0.45	-
Osteoarthritis, *n* (%)	277 (77.4)	164 (78.5)	113 (75.8)	0.61	-
Osteoporosis, *n* (%)	56 (15.6)	30 (14.4)	26 (17.4)	0.46	-
Chronic kidney disease, *n* (%)	191 (53.4)	133 (63.6)	58 (38.9)	**<0.001**	-
Urinary incontinence, *n* (%)	171 (47.9)	102 (48.8)	69 (46.6)	0.75	1
Systolic BP, mmHg, Mdn (IQR)	130 (120–140)	130 (120–140)	130 (120–140)	0.28	3
Diastolic BP, mmHg, Mdn (IQR)	70 (60–80)	70 (60–80)	70 (60–80)	0.55	3
Orthostatic hypotension, *n* (%)	50 (16.1)	27 (15.0	23 (17.7)	0.54	48
Biochemical parameters					
Osmolarity, mMol/L, Mdn (IQR)	296.5 (292.2–299.6)	298.9 (296.9–301.6)	291.1 (287.3–293.3)	**<0.001**	-
Serum sodium, mmol/L, Mdn (IQR)	140.0 (138.0–141.0)	141.0 (140.0–142.0)	138.0 (136.0–139.0)	**<0.001**	-
Hyponatremia, *n* (%)	24 (6.7)	4 (1.9)	20 (13.4)	**<0.001**	-
Serum potassium, mmol/L, M(SD)	4.43 (0.46)	4.50 (0.46)	4.33 (0.44)	**0.001**	-
Serum urea, mmol/L, Mdn (IQR)	6.9 (5.5–8.8)	7.8 (6.2–9.9)	5.8 (4.9–7.3)	**<0.001**	-
Fasting glucose, mmol/L, Mdn (IQR)	5.5 (5.0–6.3)	5.6 (5.2–6.7)	5.3 (4.9–5.8)	**<0.001**	-
>7.8mmol/L, *n* (%)	43 (12.0)	36 (17.2)	7 (4.7)	**<0.001**	
>10mmol/L, *n* (%)	12 (3.6)	10 (4.9)	2 (1.6)	0.14	
Serum calcium, mmol/L, M (SD)	1.14 (0.07)	1.15 (0.07)	1.13 (0.06)	0.14	84
eGFR, ml/min/1.73m^2^, M (SD)	57.8 (17.1)	58.2 (16.8)	58.4 (16.7)	0.91	4
Serum creatinine, mmol/L, Mdn (IQR)	86.2 (74.3–107.2)	91.05 (77.8–115.8)	78.7 (69.8–96.4)	**<0.001**	4
Hemoglobin, mmol/L, M (SD)	7.76 (1.08)	7.75 (1.08)	7.79 (1.08)	0.70	3
Anemia, *n* (%)	158 (44.5)	92 (44.7)	66 (44.3)	1.0	3
Medications					
Number of medications, Mdn (IQR)	7.0 (5.0–9.0)	7.0 (5.0–10.0)	7.0 (4.0–9.0)	**0.01**	8
Polypharmacy, *n* (%)	275 (78.6)	172 (83.5)	103 (71.5)	**0.008**	8
Alfa1-blockers, *n* (%)	21 (6.0)	13 (6.3)	8 (5.6)	0.82	9
Neuroleptics, *n* (%)	63 (18.1)	34 (16.6)	29 (20.1)	0.40	9
ACE-I/ ARB, *n* (%)	219 (62.8)	138 (67.3)	81 (56.3)	**0.04**	9
Procognitive medications, *n* (%)	45 (12.9)	20 (9.8)	25 (17.4)	0.05	9
AChE-I, *n* (%)	40 (11.5)	18 (8.8)	22 (15.3)	0.09	9
Memantine, *n* (%)	13 (3.7)	4 (2.0)	9 (6.3)	**0.046**	9
Donepezil, *n* (%)	26 (7.4)	13 (6.3)	13 (9.0)	0.41	9
Rivastigmine, *n* (%)	14 (4.0)	5 (2.4)	9 (6.3)	0.097	9
Calcium channel blockers, *n* (%)	96 (27.5)	58 (28.3)	38 (26.4)	0.72	9
Beta-blockers, *n* (%)	223 (63.9)	141 (68.8)	82 (56.9)	**0.03**	9
Digoxin, *n* (%)	27 (7.7)	17 (8.3)	10 (6.9)	0.69	9
Diuretics, *n* (%)	166 (47.6)	110 (53.7)	56 (38.9)	**0.009**	9
Lipid lowering drugs, *n* (%)	117 (33.5)	78 (38.0)	39 (27.1)	**0.04**	9
Benzodiazepines, *n* (%)	41 (11.7)	24 (11.7)	17 (11.8)	1.00	8
SSRI, *n* (%)	97 (27.8)	49 (23.9)	48 (33.3)	0.07	9
NSAIDs, *n* (%)	21 (6.0)	14 (6.8)	7 (4.9)	0.50	9

^1^ – a *t*-test for independent samples or a Mann–Whitney test (continuous or interval variables) and χ^2^ test (categorical variables) were used to compare participants categorized according to dehydration status. In all the analyses, a two-tailed *p* value of less than 0.05 was regarded as significant. ACE-I, angiotensin converting enzyme inhibitor; ARB, angiotensin II receptor blocker; AChE-I, acetylcholinesterase inhibitor; BP, blood pressure; eGFR, glomerular filtration rate; F, female; IQR, interquartile range; M, mean value; Mdn, median value; *n*, number of cases; NSAIDs, non-steroidal anti-inflammatory drugs; NYHA, New York Heart Association; SSRI, selective serotonin reuptake inhibitors; SD, standard deviation; TIA, transient ischemic attack; y, years. Bold signifies a statistically significant value.

**Table 2 nutrients-12-00398-t002:** Characteristics of participants by dehydration status- functional and nutritional parameters.

	All	Dehydration+	Dehydration-	*p* ^1^	Missing Values
*n* (%) of patients	358 (100.0)	209 (58.4)	149 (41.6)		
Functional parameters					
Barthel Index, Mdn (IQR)	90 (70–100)	90 (70–100)	90 (65–100)	0.48	2
IADL, Mdn (IQR)	7.0 (2.0–11.0)	7.0 (2.0–11.0)	7.0 (1.75–11.0)	0.55	6
AMTS, Mdn (IQR)	8.0 (6.0–9.0)	8.0 (6.0–9.0)	8.0 (6.0–9.0)	0.88	26
GDS, Mdn (IQR)	6.5 (3.0–10.0)	6.0 (3.0–10.0)	7.0 (4.0–10.0)	0.32	38
Handgrip, kg, M (SD)	18.9 (7.4)	18.93 (7.5)	18.97 (7.3)	0.96	55
man (*n* = 68)	26.3 (8.6)	26.8 (9.3)	25.7 (7.8)	0.61	16
woman (*n* = 235)	16.8 (5.5)	16.9 (5.3)	16.7 (5.7)	0.86	39
Sarcopenia, *n* (%)	140 (46.2)	80 (46.0)	60 (46.5)	1.0	55
Gait speed, m/s, M (SD)	0.68 (0.34)	0.60 (0.37)	0.69 (0.35)	0.65	88
Gait speed ≤ 0.8 m/s, *n* (%)	176 (65.2)	110 (69.2)	66 (59.5)	0.12	88
POMA score, Mdn (IQR)	23.0 (17.3–28.0)	23.0 (18.0–28.0)	23.0 (16.8–28.0)	0.92	78
POMA < 19 points, *n* (%)	81 (28.9)	47 (29.0)	34 (28.8)	1.00	78
TUG, s, Mdn (IQR)	17.4 (12.0–28.1)	18.9 (12.4–28.0)	16.6 (11.7–28.4)	0.68	97
TUG ≥ 20 s, *n* (%)	110 (42.1)	69 (45.1)	41 (38.0)	0.26	97
Falls in the last 12 months, *n* (%)	144 (46.6)	80 (44.9)	64 (48.9)	0.56	49
CFS, Mdn (IQR)	5.0 (4.0–6.0)	5.0 (4.0–6.0)	5.0 (4.0–5.5)	0.69	-
Severe frailty, *n* (%)	91 (25.4)	54 (25.8)	37 (24.8)	0.90	-
Norton scale score, Mdn (IQR)	17.0 (15.0–19.0)	17.0 (15.0–19.0)	17.0 (15.0–19.0)	0.64	2
Nutritional parameters					
BMI, kg/m^2^, M (SD)	29.0 (6.0)	29.8 (6.2)	28.1 (5.6)	**0.01**	55
BMI < 24 kg/m^2^, *n* (%)	59 (19.5)	32 (18.3)	27 (21.1)	0.56	55
BMI > 30 kg/m^2^, *n* (%)	124 (40.9)	81 (46.3)	43 (33.6)	**0.03**	55
Albumin, g/L, M (SD)	38.9 (3.8)	39.2 (3.7)	38.5 (4.0)	0.11	11
Albumin < 35 g/L	54 (15.6)	29 (14.5)	25 (17.0)	0.55	11
MNA-SF, Mdn (IQR)	11.0 (9.0–13.0)	12.0 (9.0–13.0)	11.0 (9.0–13.0)	0.26	9
MNA-SF score <8, n (%)	62 (17.8)	34 (16.7)	28 (19.2)	0.57	9
WHR, Mdn (IQR)	0.91 (0.86–0.95)	0.91 (0.85–0.96)	0.91 (0.87–0.95)	0.63	44
Waist circumference, m, Mdn (IQR)	0.97 (0.87–1.07)	0.98 (0.87–1.09)	0.94 (0.85–1.04)	**0.04**	37
Abdominal obesity, *n* (%)	271 (84.4)	166 (87.4)	105 (80.2)	0.09	37

^1^ –a *t*-test for independent samples or Mann–Whitney test (continuous or interval variables) and χ^2^ test (categorical variables) were used to compare participants categorized according to dehydration status. In all analyses, a two-tailed *p* value of less than 0.05 was regarded as significant. AMTS, Abbreviated Mental Test Score; BMI, body mass index; CFS, 7-point Clinical Frailty Scale level; GDS,15 items Geriatric Depression Scale; IADL, instrumental activities of daily living; IQR, interquartile range; M, mean value; Mdn, median value; MNA-SF, Mini Nutritional Assessment-Short Form; *n*, number of cases; POMA, Performance Oriented Mobility Assessment test; TUG, Timed Up and Go test; SD, standard deviation; WHR, waist-hip ratio. Bold signifies a statistically significant value.

**Table 3 nutrients-12-00398-t003:** Determinants of “dehydration +” status- direct multivariable logistic regression models.

	OR	95% CI	*p*	OR	95% CI	*p*	OR	95% CI	*p*
		MODEL 1			MODEL 2			MODEL 3	
Chronic kidney disease	2.40	1.3–4.1	**0.003**	2.30	1.3–4.0	**0.004**	2.27	1.3–4.1	**0.006**
Diabetes mellitus	1.80	0.96–3.4	0.07	1.75	0.9–3.3	0.08	1.75	0.9–3.3	0.09
Procognitive medications	0.48	0.2–1.1	0.08	0.37	0.1–0.99	**0.048**	0.37	0.14–0.99	**0.049**
Hypertension	1.63	0.8–3.5	0.20	1.62	0.8–3.4	0.21	1.63	0.8–3.5	0.20
Ischemic heart disease	1.12	0.6–1.99	0.69	1.12	0.6–1.99	0.70	1.11	0.6–1.98	0.72
Chronic cardiac failure	0.90	0.5–1.6	0.73	0.93	0.5–1.7	0.81	0.92	0.5–1.7	0.78
Beta-blockers	1.20	0.7–2.3	0.49	1.20	0.6–2.2	0.56	1.21	0.6–2.3	0.56
Statins	1.02	0.6–1.8	0.95	1.02	0.6–1.8	0.95	1.02	0.6–1.8	0.96
SSRI	0.80	0.4–1.5	0.50	0.79	0.4–1.5	0.49	0.80	0.4–1.5	0.50
Diuretics	1.17	0.6–2.1	0.61	1.17	0.6–2.1	0.60	1.17	0.6–2.1	0.60
ACE-I or ARB	0.91	0.5–1.8	0.78	0.93	0.5–1.8	0.83	0.93	0.5–1.8	0.84
Polypharmacy	1.12	0.5–2.4	0.78	1.16	0.5–2.5	0.72	1.16	0.5–2.5	0.72
Abdominal obesity	0.95	0.4–2.1	0.90	1.005	0.5–2.2	0.99	1.02	0.5–2.3	0.96
BMI > 30 kg/m^2^	1.13	0.6–2.1	0.69	1.12	0.6–2.0	0.72	1.13	0.6–2.1	0.71
Dementia				1.41	0.7–2.8	0.33	1.39	0.7–2.8	0.36
Age							1.01	0.96–1.05	0.84
Gender							1.05	0.5–2.1	0.88

ACE-I, angiotensin converting enzyme inhibitor; ARB-, angiotensin II receptor blocker; BMI, body mass index; CI, confidence interval; OR, odds ratio; SSRI, selective serotonin reuptake inhibitor. *p* value of less than 0.05 was regarded as significant. Bold signifies a statistically significant value.

**Table 4 nutrients-12-00398-t004:** Determinants of “dehydration+” status- stepwise backward multivariable logistic regression analysis.

	OR	95% CI	*p*
		MODEL 4	
Chronic kidney disease	2.53	1.51–4.27	**<0.001**
Diabetes mellitus	1.9	1.04–3.44	**0.04**
Procognitive medications	0.42	0.20–0.88	**0.02**
Hypertension	1.75	0.91–3.34	0.09

Variables included in the analysis: hypertension, ischemic heart disease, cardiac heart failure, diabetes/ prediabetes, chronic kidney disease, taking procognitive medications, beta-blockers, statins, selective serotonin reuptake inhibitors (SSRI), diuretics, ACE-I or ARB, abdominal obesity, BMI > 30 kg/m^2^, polypharmacy, dementia, age, and gender. *p* value of less than 0.05 was regarded as significant. Bold signifies a statistically significant value.

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
