# Peer review of "Impending Low Intake Dehydration at Admission to A Geriatric Ward- Prevalence and Correlates in a Cross-Sectional Study"

_nutrients, 2020, doi:10.3390/nu12020398_

Round 1
Reviewer 1 Report
This by ZB Wojszel is a well-conducted cross-sectional study which aims to describe prevalence and characteristics of dehydrated patients at admission to a Polish geriatric ward. Although the study idea is not completely original (Hooper et al. J Gerontol A Biol Sci Med Sci. 2016) and the results are limited by the study design, the study is well conducted and reported and it contributes to stress the importance of inadequate hydration in the geriatric population and it helps recognizing this widespread clinical issue.
The strength of this study lies in the very detailed description of many sociodemographic and health characteristics of a good number of consecutively admitted-to-hospital geriatric patients. The work is well organized and described, and the paper is easy to read.
I hereby provide three major comments possibly addressing the few weaknesses of this study:
The distinction between “calculated” (or “counted”) osmolarity and “measured” osmolality is not always clear throughout the text. Please specify when necessary, prefer “calculated” instead of “counted”, and be consistent. The same applies for “impending dehydration” (diagnosed if calculated osmolarity > 295 mMol/L) and true “dehydration” (diagnosed if measured osmolality > 301+/-5 mMol/Kg). Try to be clear from the beginning (see lines 286-289), specify when necessary to avoid confusion, and be consistent through the text using always the same terms. As an example, from lines 135-136, if in 58.4% of patients the calculated osmolarity exceeded the threshold of 295, than 292.9 could not possibly be the median value of the same set of values. If I am not wrong, a clear distinction between these two sets of values is missing. Does the latter refer to measured osmolality? Following the previous comment, if two distinct set of values (for calculated and measured osmolarity) are available, this should be clearly represented in Table 1. Moreover, a measure of correlation between calculated and measured dehydration could be interesting to have in the text and maybe to also present in a simple figure (eg, a scatter plot). The independent association between the variables (covariates) and the impending dehydration (outcome) is correctly assessed with a multivariable logistic regression model. Although difficult to say without the original dataset, the presented model could (possibly) suffer from overfitting. In fact, the inclusion of too many variables could lead to loss of generalizability of the prediction model, and maybe even reduce its prediction ability. The Author included in the final analysis significant (P<0.1) predictors at mean/median comparison. Although this is a common strategy for constructing logistic regression models, it is not mandatory to include all of them. Using simple model building strategies, such as forward selection or backward elimination, the Author could try and fit some more parsimonious models and test their predictive performance. In particular, I suggest the Author to adjust for biologically important covariates such as sex and age (even if not significant at crude comparison), and to avoid covariates with lack of biological plausibility (by common sense) and/or with many missing values (such as BMI and abdominal obesity) since they lead to reduction of the sample size, loss of generalizability and possibly bias. If a similar strategy for model building has already been used, please describe it in the “Material and Methods” section and overlook this comment.Also, I suggest to include in the introduction or discussion the work by Białecka-Dębek A and Pietruszka B, Aging Clin Exp Res. 2019 (doi: 10.1007/s40520-018-1019-5).
Finally, I hereby list a few minor comments and suggestions:
I suggest not listing all the multivariable covariates (lines 20-23). These could be summarized with “health, biochemical, and nutritional parameters and medications”. The Author could consider using high/low (odds/risk) instead of positive/negative (trend/effect) - line 19 and throughout the text - for clarity (eg, the one reported as “positive effect” is negative = increasing the risk of dehydration). Line 125. Please specify the significant P value for statistical tests comparing means and medians, both in the text and tables. Line 151. Please check the percentages: 53.3% vs 56.3% in Table 1. Line 171. Please verify if “WHR” and “percentage of patients with abdominal obesity” are significant parameters (P > 0.05 in Table 2). Lines 226-227. To better assess the role of age I suggest to include this variable (together with sex) in the multivariable logistic regression model (see major comment above). Figure 1. At “patients not eligible” please use “and/or” for accuracy. Also, please prefer calculated instead of counted. Table 1: Please check the table’s caption, it should include “biochemical parameters” and “medications”. Table 1 and Table 2: Please prefer “Mdn” instead of “Me” for median, please revise the tables accordingly. Table 1, 2 and 3: Please highlight significant values.Author Response
Please see the attachment.

Reviewer 2 Report
This is an interesting study, very important in clinical practice. However, I have some important comments.
In table I what means cardiac heart failure in terms of class NYHA? III and IV class NYHA could have some degree of congestion (or hyper hydration). It is very important to know the percentages of these classes of heart failure in each group even P is non-significantly.
Treatment with diuretics is also a problem for this subject. In my opinion, these patients must to be excluded.
Non-assessing plasma osmolality is a major issue of this study.
What percentage of patients with chronic kidney disease have been under treatment with diuretics? This is another major issue for the conclusion of this study.
This study is about frailty in geriatric patients. The Ethic Committee of the hospital approved it?
Round 2
Reviewer 1 Report
I really appreciate the thorough work of the Author for positively addressing all my comments.
After revising the new version of the manuscript, I hereby provide just a few minor suggestions:
1) In the Title, I suggest adding "a" cross-sectional study.
2) Line 9, I suggest "This" cross-sectional study [...].
3) Lines 14-16, only the results from Table 3 (Model 1-3) should be reported as controlled for several parameters. "Significantly higher odds for impending dehydration" were observed only for CKD with trends for diabetes and procognitive medication when "controlling for several [...] parameters".
4) Line 230, Model "4" instead of Model 3.
5) Lines 281-283, the Author should highlight that, based on Model 4, CKD, DM, taking procognitive medications and hypertension are the main variables for the outcome prediction.
Reviewer 2 Report
Thank you for your answers.
